# Rapid Diagnosis and Visual Detection of Potato Cyst Nematode (*Globodera rostochiensis*) Using Recombinase Polymerase Amplification Combination with Lateral Flow Assay Method (RPA-LFA)

**Xu Wang [1,2], Rong Lei [2], Huan Peng [1], Ru Jiang [1], Hudie Shao [1], Jianjun Ge [2,*] and Deliang Peng [1,*]**

1   State Key Laboratory for Biology of Plant Diseases and Insect Pests, Institute of Plant Protection, Chinese Academy of Agricultural Sciences, Beijing 100193, China
2   Institute of Plant Quarantine, Chinese Academy of Inspection and Quarantine, Beijing 100176, China
*   Correspondence: jianjun11@163.com (J.J.G.); dlpeng@ippcaas.cn (D.L.P.);
    Tel.: +86-10-53897649 (J.J.G.); +86-10-62815576 (D.L.P.)

**Abstract:** *Globodera rostochiensis* is an important quarantine pest, it causes serious potato yield losses annually. Reliable and rapid molecular detection of *G. rostochiensis* is pivotal to effective early disease diagnosis and managements. Herein, recombinase polymerase amplification integrated with lateral flow assays method (RPA-LFA) was developed to target the internal transcribed spacer of nuclear ribosomal DNA (ITS rDNA) of the golden cyst nematode (*G. rostochiensis*), which allowed for the rapid diagnosis and detection of this nematode from crude extracts of cysts and juveniles within 30 min. Sensitivity test results showed that $10^{-1}$ single juvenile and $10^{-3}$ single cyst can be reliably detected. Moreover, the RPA-LFA method can directly diagnose and detect *G. rostochiensis* from infested field soil. This is the first RPA-LFA method for diagnosis *G. rostochiensis*, it is a fast, accurate, and sensitive detection method and can be developed for detection of *G. rostochiensis* in fields and laboratories lacking large instrument and equipment.

**Keywords:** golden potato cyst nematode; *Globodera rostochiensis*; recombinase polymerase amplification; lateral flow assays

## 1. Introduction

Plant-parasitic nematodes (PPNs) cause a worldwide economic loss of more than USD 157 billion per annum [1]. The root-knot nematodes (*Meloidogyne* spp.) and the cyst nematodes (*Heterodera* spp. and *Globodera* spp.) are economically the most damaging of all the PPNs [2]. The potato cyst nematodes (PCNs), including *Globodera rostochiensis* and *Globodera pallida*, are very important quarantine nematodes in potato fields [3,4], which are subjected to strict quarantine regulations in many countries [5]. Both species cause serious potato yield losses. It can reduce potato yield by 9% in Europe and other areas of the world without taking any control strategies [6]. The high PCN populations in the field can cause potato yield losses up to 80~90%, or even total crop failure [7]. General symptoms on potato field infested by PCN are stunted growth, yellowing leaves, and reduced number of tubers and their size [8]. These symptoms are not specific, so it is difficult to diagnosis PCN at the initial stages of infection. Furthermore, PCN eggs can survive in a dormant stage inside the cyst for more than 18 years [9], implying the huge challenge to eradicate the pest once soils are infested [8,10]. To prevent the further spread of PCN, strict quarantine regulations in many countries and areas have been implemented [11,12]. *G. rostochiensis* was reported in Sichuan, Yunnan, and Guizhou provinces in China in 2022 [13,14].

The initial stage of detection of PCN is critical to adopt a timely management strategy [6]. PCN species are usually distinguished through morphology of the cysts, juveniles,

and males. It requires a large number of samples, a skilled nematologist and time consumption [15]. *G. rostochiensis* and *G. pallida* are morphologically and morphometrically similar [6], so a combination of morphological characters and molecular techniques makes the identification of PCN in different growth stages more accurate. Molecular techniques are recommended in European and Mediterranean Plant Protection Organization (EPPO) for routine testing [16]. In recent years, PCR-based methods have been reported for identification and diagnosis of *G. rostochiensis* using conventional PCR [17–21], and TaqMan real-time PCR [22–25]. DNA barcoding based on cytochrome c oxidase subunit I (COI), 18S rDNA, and 28S rDNA can support the identification of *G. rostochiensis* [26]. However, with the development of isothermal amplification technology, the shortcomings of conventional molecular methods become more and more obvious. These methods either take a long time for PCR reaction and PCR product detection or require large expensive instruments and equipment. Recently, a loop-mediated isothermal amplification (LAMP) assay using five primers was also developed to identify *G. rostochiensis* [27].

At present, isothermal nucleic acid amplification techniques are alternative molecular techniques for species identification. The most common isothermal amplification technique includes LAMP, cross primer amplification (CPA), and recombinase polymerase amplification (RPA). The temperature required for LAMP amplification is 68 °C [27]. RPA reaction uses three highly efficient enzymes that can be realized at a constant temperature between 37 and 42 °C, and the template DNA is exponentially amplified in 30 min, so it is a good choice to develop on-site rapid detection methods [28]. RPA product detection is mostly used by agarose gel electrophoresis or combined with fluorescent probes by real-time PCR or using lateral flow dipstick. At present, many researchers have used the RPA to detect plant pathogens, such as fungi, bacteria, and viruses [29–32]. RPA assays have been used to diagnose a variety of plant-parasitic nematodes including root-knot nematodes such as *Meloidogyne javanica*, *M. arenaria*, *M. incognita*, *M. hapla*, and *Bursaphelenchus xylophilus* [33–38]. In addition, RPA combined with CRISPR was developed for the detection of *Heterodera schachtii* [39]. Until now, RPA has yet to be applied for detection of potato golden cyst nematode *G. rostochiensis*.

In this study, we developed RPA combined with lateral flow assay (RPA-LFA) for the detection of *G. rostochiensis* using the crude extracts and in field soil. The advantages of this assay are rapid reaction, simple operation, and visualization of results. Compared with traditional detection methods, the RPA-LFA method is more suitable for the fast diagnostics of *G. rostochiensis* in field and at the import and export port.

## 2. Materials and Methods

### 2.1. Nematode Populations

Nine populations of *Globodera rostochiensis* from Guizhou, Yunnan, and Sichuan Provinces (China) and Norway, one population of *G. pallida* from Belgium, one population of *G. artemisiae* from Guizhou Province (China), and three other cyst nematode species were used in our work (Table 1). All nematode populations were examined using morphological identification and ITS ribosomal DNA (rDNA) sequence. Second-stage juveniles (J2s) were collected from the broken cysts.

### 2.2. DNA Extraction

The crude nematode DNA was extracted from a single cyst or from a single second-stage juveniles (J2). One cyst was transferred into a 200 μL PCR tube containing 10 μL lysis solution A and 2 μL lysis solution B (Ningbo Zhenhai, Baichuan Biotechnology, Wenzhou, China). The cyst was broken using a pipette tip. A single J2 was placed in 10 μL lysis solution A on a microscope slide under a stereo microscope and cut into two sections with an ophthalmic scalpel. A pipette was used to suck the mixed solution into a PCR tube and add lysis solution B. Then, these PCR tubes containing nematode and solution mixture were incubated at 95 °C for 10 min, followed by transient centrifugation. The crude DNA preparation was diluted using 20 μL nuclease-free water and stored at −20 °C until further

use. To isolate nematode DNA from natural field soil (Table 2) up to 500 mg of soil was added to the Lysing Matrix E Tube. A total of 978 μL Sodium Phosphate Buffer and 122 μL MT Buffer were added. The tubes vortex was secured to mix for 40~50 s. Then, total genomic DNA was extracted using the FastDNA^TM SPIN Kit (No. 116560200, MPBIO, Irvine, CA, USA), according to the manufacturer's protocol. DNA was stored at −20 °C until further use.

**Table 1.** Populations of cyst nematode used to determine the specificity of the recombinase polymerase amplification integrated with lateral flow assay (RPA-LFA).

| Code | Species | Host | Origin |
|---|---|---|---|
| Gr1 | | | Hezhang county, Guizhou, China |
| Gr2 | | | Hezhang county, Guizhou, China |
| Gr3 | | | Weining, Guizhou, China |
| Gr4 | | | Ludian county, Yunnan, China |
| Gr5 | *G.rostochiensis* | Potato | Zhaojue county, Sichuan, China |
| Gr6 | | | Norway |
| Gr7 | | | Norway |
| Gr8 | | | Norway |
| Gr9 | | | Norway |
| Ga | *G.artemisiae* | Artemisia argyi | Guizhou, China |
| Gp | *G.pallida* | Potato | Belgium |
| Ce | *Cactodera estonica* | Potato | Neimenggu, China |
| Hs | *H.schachtii* | Sugar beet | Xinyuan county, Xinjiang, China |
| Hg | *H.glycines* | Soybean | Langfang, Hebei, China |

**Table 2.** Soil samples collected from nine locations for detection of detection of *G. rostochiensis* RPA-LFA and the conventional PCR assay.

| Code | Host | Location | Sampling Date |
|---|---|---|---|
| 1 | | Hezhang county, Guizhou, China | August 2020 |
| 2 | | Hezhang county, Guizhou, China | August 2020 |
| 3 | | Weining, Guizhou, China | August 2020 |
| 4 | | Ludian county, Yunnan, China | September 2020 |
| 5 | Potato | Yunnan, China | September 2020 |
| 6 | | Zhaojue county, Sichuan, China | September 2020 |
| 7 | | Sichuan, China | September 2020 |
| 8 | | Huludao, Liaoning, China | September 2020 |
| 9 | | Linyi, Shandong, China | July 2020 |

*2.3. RPA Primer and Probe Design and Testing*

ITS-rDNA region sequences of *Globodera* spp. (including *G. rostochiensis* EU517119, FJ212166, GQ294521, JF907541, KJ409617; *G. pallida* EU855119, FJ212165, HQ670280, JQ692594, KY660056; *G.artemisiae* AF161003, AF274415, AY519126, EU855121, EU935420) and other cyst nematode (*Cactodera estonica* HM560730; *Heterodera schachtii* EF611123; *H.glycines* KP324915) were retrieved from GenBank (http://www.ncbi.nlm.nih.gov, accessed on 6 May 2020). These sequences (Table 3) were aligned using DNAMAN 8.0 Software (Lynnon Biosoft, Vaudreuil, QC, Canada). Six pairs of RPA primer were designed (28–35 bp) according to the sequence variance between the ITS rDNA region of these cyst nematodes. The amplification of six pairs of primers were verified by RPA Basic kit (TwistDx, Cambridge, UK). According to the manufacturer's instruction, we prepared the reaction system, removal of 29.5 μL supplied the rehydration buffer, 2.4 μL 10 μM forward primer, 2.4 μL 10 μM reverse primer, 2 μL DNA template, and 11.2 μL distilled water into the lyophilized reaction pellets, which were vortexed and mixed, followed by the addition of 2.5 μL of magnesium acetate (280 mM) to initiate the RPA reaction at 39 °C in Mini Metal Bath for 30 min. Amplification products were detected by agarose gen electrophoresis. The probe for lateral flow assay was modified with Fam group and tetrahydrofuran (THF) group and was cleaved by the

Nfo endonucleases to generate extensible 3′-OH group for polymerization. The generated products combined with the Biotin-labelled reverse primer led to the production of a dual-labelled amplicon.

### 2.4. RPA-LFA Reaction

A master-mix consisting of 29.5 μL supplied rehydration buffer, 12.2 μL nuclease free water, 2.1 μL 10 μM forward primer, 2.1 μL 10 μM reverse primer, and 0.6 μL 10 μM probe were added to the RPA TwistAmp^(TM) nfo enzyme pellet (TwistDx, Cambridge, UK) to rehydrate the enzymes. Then, 2 μL of target DNA were added into the RPA buffer, followed by the addition of 2.5 μL of magnesium acetate (280 mM) to initiate the RPA reaction at 39 °C in Mini Metal Bath for 30 min. After the RPA reaction, 5 μL of each product was transferred into a 1.5 mL centrifuge tube where 95 μL of Dipstick Assay Buffer (Milenia Biotec GmbH, Giessen, Germany) was loaded, then an LFD strip was inserted into the mixed solution of the centrifuge tube. The results were observed after 5 min by the naked eye. The positive result showed both test and control lines, whereas only one control line was observed for the negative samples.

### 2.4.1. Specificity Test

To determine the specificity of the RPA-LFA, the single cyst crude extracts from 14 cyst nematode populations (including nine *G. rostochiensis* from different geographical origins, one *G. pallida*, one *G. artemisiae*, and three other cyst nematodes species) were used as DNA templates for amplification by RPA-LFA. All nematode templates were amplified by PCR using the specific primers ITS5, PITSr3, and PITSp4 of PCN to confirm the nematode species. The length of the amplified fragment is 434 bp, which can be confirmed as *G. rostochiensis*. Sterilized distilled water was used for the negative control (NTC) reaction. The test was three replicates for each sample.

### 2.4.2. Sensitivity Test

To evaluate the sensitivity of the RPA, ten-fold serial diluted DNA (1, 1/10, 1/100, 1/1000 per reaction tube) isolated from *G. rostochiensis* were used as a standard to determine sensitivity and obtain the amplification efficiency. The crude DNA extracted from one cyst or single juvenile nematode was diluted with sterile water and used as the initial DNA. DNA of different dilutions described above were analyzed separately using the RPA method. The sensitivity test was three replicates for each treatment.

### 2.5. Detection of Globodera rostochiensis from Actual Field Soil

To determine the ability of the RPA-LFA to detect *G. rostochiensis* from the soil, three potato rhizosphere soil samples were collected from potato cultivation areas in Guizhou Provinces during the potato planting period. Guizhou Province is the area where potato cyst nematodes were first reported in China. Six other soil samples were collected from potato plants in Sichuan, Yunnan, Liaoning, and Shandong Provinces (China). All soil samples were collected in 2020 (Table 2). The Cobb decanting and sieving method was used to isolate cysts from the 50 g soil samples. Isolation, counting, and morphological identification of cysts were performed under light microscope. DNA was directly extracted by the FastDNA^(TM) SPIN Kit for soil (No. 116560200, MPBIO, Irvine, CA, USA), and detected with the RPA-LFA method. To evaluate the accuracy of the detection results, the test was also performed using conventional PCR with the previously reported *G. rostochiensis* the specific primers ITS5 and PITSr3 [19]. PCR amplification was carried out with the universal primer D2A and D3B of the nematode to confirm that the nucleic acid of the nematode was successfully extracted from each isolate. *G. rostochiensis* genomic DNA was used as a positive control and non-infested soil as a negative control. The test was repeated three times for each sample.

## 3. Results

### 3.1. Primer and Probe Design

Several sequences of the ITS rDNA for *G. rostochiensis* and other nematodes were retrieved from the Genbank and aligned with the ClustalW algorithm in DNAMAN. Several sets of species-specific *G. rostochiensis* primers were designed based on the regions with high dissimilarity among species.

### 3.2. RPA Detection

Six pairs of candidate primers were used to amplify the DNA of *G. rostochiensis* by RPA Basic Kits, and the products were detected by agarose gel electrophoresis. The amplification fragment length approximately 265 bp (Figure 1) and confirmed by sequencing. Considering the brightness and length of the product, and the primer dimers, set 4 primers (GF4 and GR4) were selected for subsequent experiments. The final sequences of primers and probe used for the assays are listed in Table 3. The specificity of primers and probes in sequence alignment results of several nematodes is shown in Figure 2.

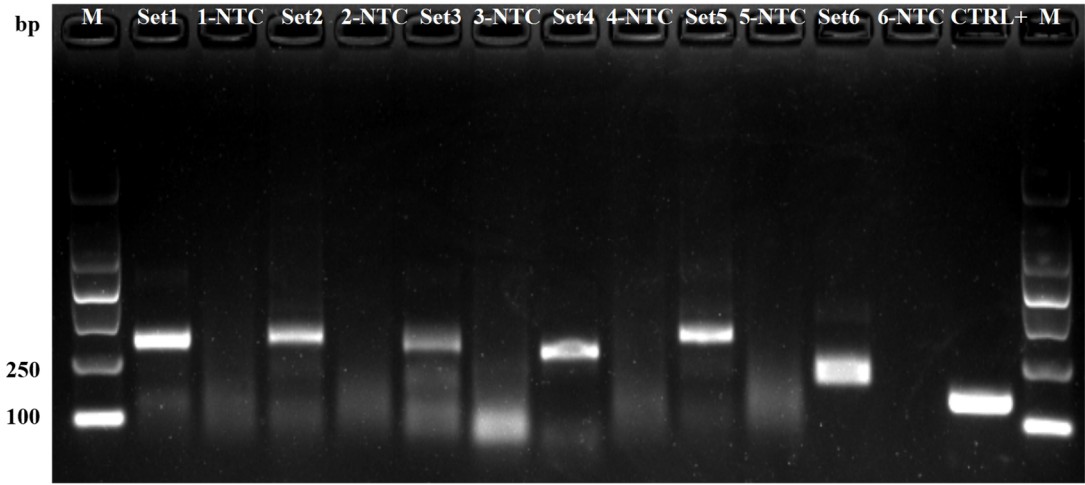

**Figure 1.** RPA amplicon of the partial ITS rDNA gene sequences on agarose gel. Lanes: M: DL2000 DNA marker; Set 1~Set 6: RPA amplicon obtained with six sets of primers using TwistAmp Basic kit; 1-NTC~6-NTC: no template control with six primers sets; CTRL+: the positive control provided by the kit.

**Table 3.** Sequences of the primers and a probe used in *G. rostochiensis* RPA-LFA and conventional PCR assay.

| Name | Sequences (5′–3′) | Usage | Reference |
|---|---|---|---|
| GrF4 | CTGTGTATGGGCTGGCACATTGACCAACA | | |
| GrR4 | [Biotin] TACGGCACGTACAACATGGAGTAGCAGCTAC | *G. rostochiensis*-specific RPA-LFA primers and probe | This study |
| GrP | [Fam] [1] CGGAGGAAGCACGCCCACAGGGCACCCTAACG [THF] [2] CTGTGCTGGCGTCTGT [C3-spacer] [3] | | |
| ITS5 | GGAAGTAAAAGTCGTAACAAGG | | |
| PITSr3 | AGCGCAGACATGCCGCAA | *G. rostochiensis*-specific PCR primers | [19] |
| PITSp4 | ACAACAGCAATCGTCGAG | | |
| D2A | ACAAGTACCGTGAGGGAAAGTTG | 28S rDNA universal PCR primers | [40] |
| D3B | TCGGAAGGAACCAGCTACTA | | |

[1] Fam—fluorophore: [2] THF—tetrahydrofuran: [3] C3—spacer block.

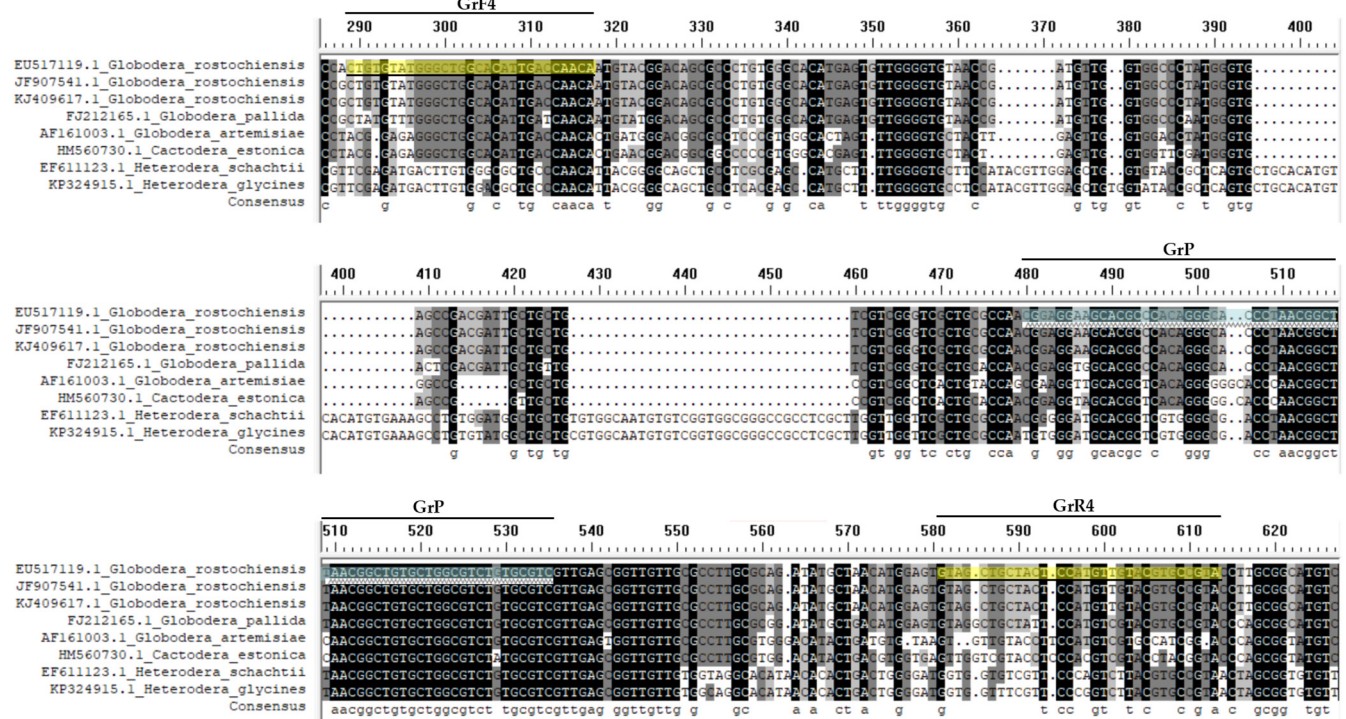

**Figure 2.** The fragment of alignment of the ITS rDNA for several cyst nematodes, showing the positions of RPA-LFA primers (GrF4 and GrR4, in yellow) and probe (GrP, in blue) used in this study.

### 3.3. Specificity Test

The specificity of the RPA-LFA was evaluated using nine populations of *G. rostochiensis* and five other nematode species including: *G. pallida*, *G. artemisiae*, *C. estonica*, *H. schachtii*, and *H. glycines* (Table 1). The RPA-LFA result showed high level specificity to *G. rostochiensis*. Only the strip of *G. rostochiensis* obviously showed both test line and control line, but the other cyst nematode species and negative control showed no test line (Figure 3A). The results of RPA-LFA were consistent with those of normal PCR using ITS5, PITSr3, and PITSp4 primers (Figure 3B). These results indicated that the RPA-LFA can specifically distinguish *G. rostochiensis* from other nematodes. The triplicate samples for each *G. rostochiensis* population all obtained positive results.

### 3.4. Sensitivity Test

The sensitivity of RPA-LFA were evaluated with diluted crude DNA extracted from a single cyst and a single juvenile of *G. rostochiensis* (Figure 4). The results showed that the RPA-LFA assays can obviously detect $10^{-1}$ dilution of DNA extracted from a single juvenile and $10^{-3}$ dilution of DNA extracted from a single cyst. The same results were obtained for the triplicate.

### 3.5. Detection of Globodera rostochiensis in Actual Field Samples

Nine soil samples from potato rhizosphere soil listed in Table 3 were used to determine the practical application of the RPA-LFA. As shown in Table 4 and Figure 5, seven soil samples (Guizhou, Sichuan, and Yunnan Province) showed positive results using RPA-LFA technologies, while the other two samples were negative (Figure 5A). The results were consistent with conventional PCR using specificity primers ITS5/PITSr3 (Figure 5B). All samples were amplified successfully with nematode universal primers D2A/D3B (Figure 5C). This indicated that all soil samples contain nematodes. The triplicate samples for each *G. rostochiensis* population all obtained positive results.

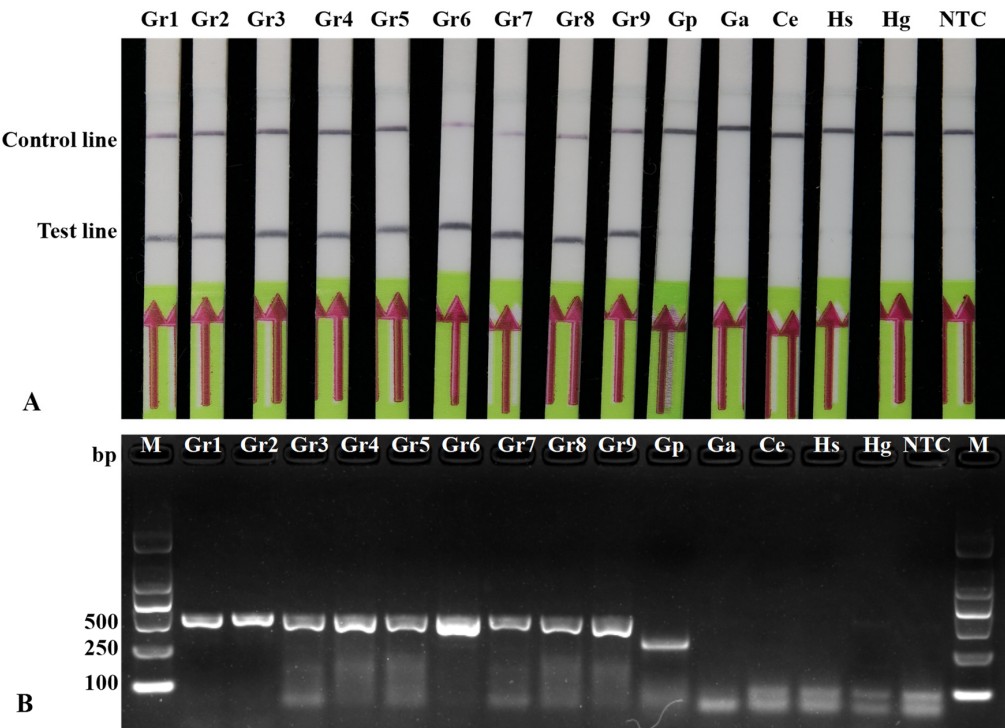

**Figure 3.** Verification of assay specificity for *G. rostochiensis*. Gr1–Gr9: *G. rostochiensis* from Guizhou, Yunnan, and Sichuan Provinces in China and Norway, Gp: *G. pallida* from Belgium, Ga: *G. artemisiae* from Guizhou Province in China, Ce: *C. estonica* from Neimenggu Province in China, Hs: *H. schachtii* from Xinjiang in China, Hg: *H. glycines* from Hebei in China. (**A**) RPA-LFA test results. The presence of two line indicates a positive result. (**B**) PCR results with primers ITS5, PITSr3, and PITSp4. M: DL2000 marker; NTC: no template was added as a negative control.

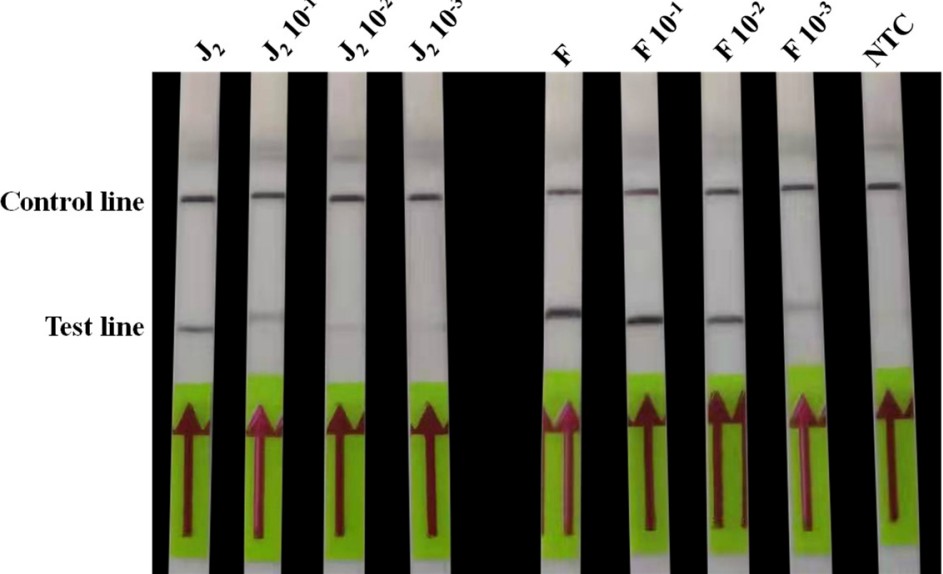

**Figure 4.** Verification of sensitivity of the RPA-LFA using dilutions of DNA crude extract of *G. rostochiensis*. J$_2$: second-stage juvenile; F: cyst; NTC: no template was added as a negative control; the presence of two line indicates a positive result.

**Table 4.** Detection of *G. rostochiensis* in soil samples using the RPA-LFA and the conventional PCR assay.

| Code | Density of Cysts [1] | Detection Results | | |
|---|---|---|---|---|
| | | RPA-LFA | PCR (ITS5/PITSr3) | PCR (D2A/D3B) |
| 1 | 437 ± 28 | + | + | + |
| 2 | 344 ± 26 | + | + | + |
| 3 | 542 ± 36 | + | + | + |
| 4 | 280 ± 18 | + | + | + |
| 5 | 251 ± 7 | + | + | + |
| 6 | 379 ± 29 | + | + | + |
| 7 | 361 ± 16 | + | + | + |
| 8 | 0 | − | − | + |
| 9 | 0 | − | − | + |

[1] Density of cysts were the number cysts of *G. rostochiensis* in 50 g soil.

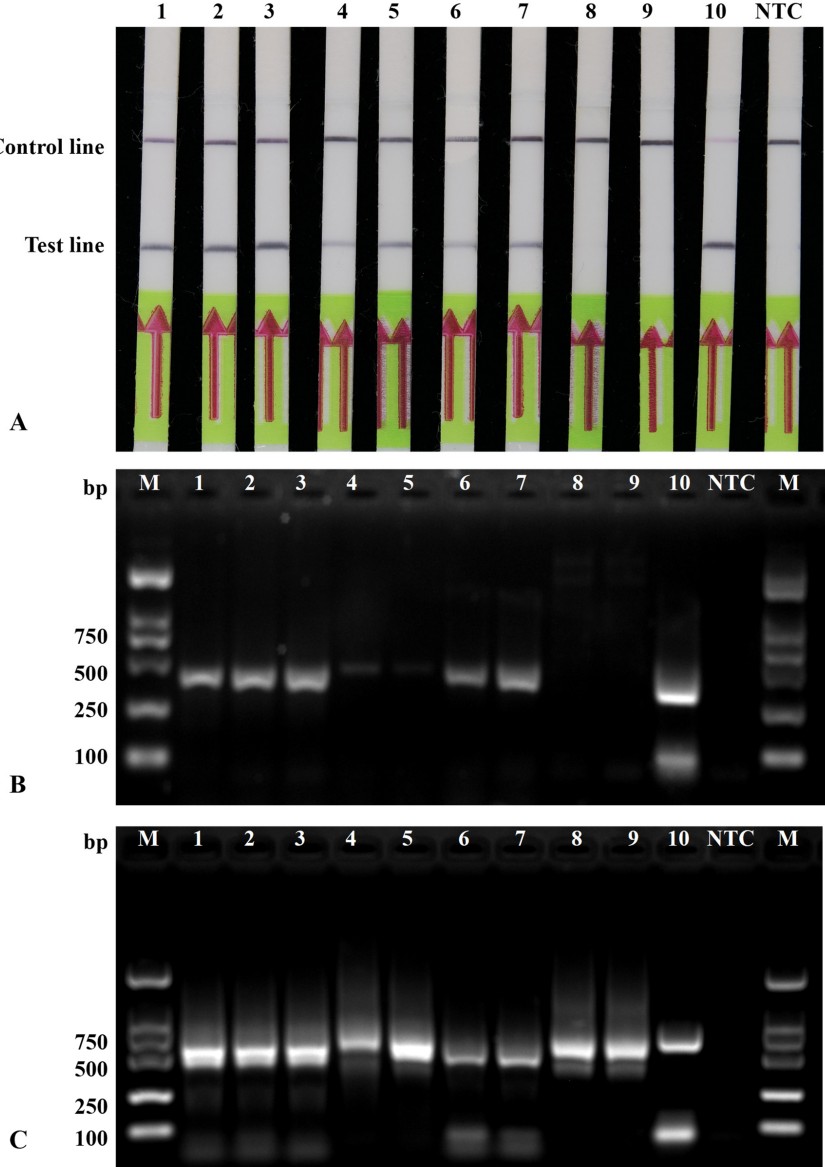

**Figure 5.** Detection of *G. rostochiensis* in soil samples (Table 2) using the PCR assays and RPA assay. (**A**) RPA-LFA results. The presence of two line indicates a positive result; (**B**) PCR results with primers ITS5/PITSr3. (**C**) Conventional PCR results with primers $D_2A/D_3B$. M: DL2000 marker; 10: the positive RPA-LFA was DNA of *G. rostochiensis*; NTC: no template was added as a negative control.

## 4. Discussion

The golden nematode was recently reported from Guizhou, Yunnan, and Sichuan Provinces of China [13,14]. Early and timely detection of PCN in the field or in the process of quarantine surveys is essential for their control. Morphological diagnosis and previous PCR detection assays are considered the standard detection methods. However, these methods are only available in laboratories with microscopes and thermal cycling equipment. To detect *G. rostochiensis* in field, we developed a simple, fast, and visual RPA-LFA, which requires a mini portable heater and pipettes.

The ITS rDNA was selected for the RPA-LFA because previously published PCR assays proved its feasibility of distinguishing *Globodera* species [41,42]. Based on the ITS rDNA fragment of the PCN, many scholars have designed several pairs of specific primers and specific probes using PCR, multiplex PCR, real-time PCR, multiplex real-time PCR, etc., to diagnose *G. rostochiensis* [17,19,22,25]. Using the ITS rDNA gene as the target, *G. rostochiensis* can be distinguished from the other cyst nematode species, including *Globodera* spp. and *Heterodera* spp. using the RPA-LFA, showing high specificity.

The RPA-LFA developed in this study allowed the detection of the juveniles and cyst life stages with a crude extraction kit. A simple and rapid DNA extraction method is the first step to realize the rapid detection in the field. Two methods of nucleic acid extraction from nematodes are often used at present. The first one is suitable to extract nucleic acid of single nematode. The nucleic acid of single nematode is extracted using PCR buffer and proteinase K after heating. Nematode nucleic acid can be obtained by heating the nematode in the mixture of PCR buffer and proteinase K [33,36]. The kit extraction method is the main method for extracting genomic DNA of nematodes from plants and other samples [35]. The rapid crude extraction kit used in the present study needs only one step. Therefore, the entire detection process for the RPA-LFA takes approximately 46 min to complete, including 10 min for crude extract DNA, 30 min for the RPA reaction, less than 1 min to dilute the reaction product, and 5 min for visual detection using the LF strips. The reaction time of RPA is 20–30 min, so rapid nucleic acid extraction is the key to shortening the time of the molecular detection process [36,43].

The soil samples collected in our study were from potato rhizosphere soils during potato growth. The soil contained a large number of cysts. *G. rostochiensis* can be successfully detected by RPA-LF assay. PCN mainly survives in the soil for a long time in cysts, for which hatching is less than 30% in the absence of a host plant [44]. Therefore, reproducibility of the assays in extracts from soil samples with only few old cysts should be further tested to clarify the stability of RPA reactions.

The results showed that the sensitivity of the RPA-LFA method is similar to that using a PCR species-specific primer. However, compared with PCR-based methods, RPA-LFA has several important advantages. The first one is that RPA-LFA results can be observed in 1 h, whereas the reaction time of PCR assays more than 2 h. The second advantage is that the result of RPA-LFA was visible to the naked eye, whereas PCR assays required visualization of results by gel electrophoresis techniques. Thus, RPA-LFA is a promising alternative to PCR methods for rapid detection of nematodes.

## 5. Conclusions

In the present study, the RPA-LFA diagnosis and detection method for *G. rostochiensis* was developed. The method has high specificity and sensitivity. It can rapidly, visually, accurately, and directly detect *G. rostochiensis* from infested soil. The application of the RPA-LFA assay has great potential for diagnosing infestations of *G. rostochiensis* in the lab or in the field with minimal laboratory equipment and rudimentary experimental conditions.

**Author Contributions:** Conceptualization, X.W. and R.L.; methodology, X.W.; software, R.L.; validation, X.W., R.L. and H.P.; formal analysis, J.J.G.; investigation, R.J. and H.D.S.; resources, D.L.P.; data curation, X.W.; writing—original draft preparation, X.W.; writing—review and editing, R.L.; visualization, D.L.P.; supervision, D.L.P.; project administration, D.L.P.; funding acquisition, D.L.P. All authors have read and agreed to the published version of the manuscript.

**Funding:** This research was supported by the Sichuan Science and Technology Program (2021YFN0009), the National Key R&D Program of China (2021YFD1400100), the National Natural Science Foundation of China (32072398), and the Science and Technology Innovation Project of Chinese Academy of Agricultural Sciences (ASTIP-02-IPP-15), Basic Scientific Research Foundation of Chinese Academy of Inspection and Quarantine (2020JK018).

**Institutional Review Board Statement:** Not applicable.

**Informed Consent Statement:** Not applicable.

**Data Availability Statement:** The datasets generated during and/or analyzed during the current study are available from the corresponding author on reasonable request.

**Acknowledgments:** The authors are grateful to Mingyi Feng, the Plant Protection Station of Hezhang of Guizhou Province, Bingzhi Xu at Yuexi Plant Protection Station of Sichuan Province and Youhua Chen at Zhaojue Plant Protection Station of Sichuan Province for collecting and sending the samples to Institute of Plant Protection, Chinese Academy of Agricultural Sciences in August 2019. We thank Richard Smiley (Oregon State University) for reading and editing this manuscript.

**Conflicts of Interest:** The authors declare no conflict of interest.

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
