# Peer review of "Rapid Diagnosis and Visual Detection of Potato Cyst Nematode (Globodera rostochiensis) Using Recombinase Polymerase Amplification Combination with Lateral Flow Assay Method (RPA-LFA)"

_agronomy, doi:10.3390/agronomy12102580_

Round 1

Reviewer 1 Report

The manuscript present valuable research on the development of a rapid and reliable assay for the detection of G. rostochiensis in soil. This type of diagnostic is of importance in areas where the nematode is found and extensive laboratory infrastructure does not exist. It appears that all of the right steps were taken to develop and validate the methodology. However, lack of detail in the methods section makes it difficult to fully evaluate the work. Specific comments include:

Specificity test: What life stage of the nematode was used in these assays? Of the triplicate samples for each population did all result in the expected result (either positive or negative for G. rostochiensis)?

Sensitivity test: A 10-fold dilution of DNA from J2 and cysts was prepared, but only a 3-fold dilution is shown. Please discuss the rest of the data.

Application to soil: This section needs much more information. Include soil types for each location. What was the population density of cysts/J2 in these samples? Where there cysts and J2 in all of the samples? How DNA was extracted from soil and volume of soil used must be included and described in detail. Mention of whether all three replicates resulted in the same result needs to be mentioned. If the opportunity exists, then the replication should be increased to 10 to demonstrate reproducibility of the method soil samples – this will make the results much stronger.

In the results and in a figure mention results from D2A/D3B – however, this is not mentioned in the methods. Make sure the reader knows these primers were used and why.

Why is the conclusion before the discussion?

Statement comparing difficulty of using ITS vs. effector gene needs more context in discussion.

It is state in discussion that cysts and J2 were in soil samples, but no evidence of this in methods or results.

Author Response

Response to Reviewer 1 Comments

Point 1: Specificity test: What life stage of the nematode was used in these assays? Of the triplicate samples for each population did all result in the expected result (either positive or negative for G. rostochiensis)? 

Response 1: We are grateful to the reviewer for this suggestion. We have redefined this section in the revised manuscript. The single cyst crude extracts were used in specificity test. The same results were obtained for the triplicate, and all G.rostochiensis samples were positive.

Point 2: Sensitivity test: A 10-fold dilution of DNA from J2 and cysts was prepared, but only a 3-fold dilution is shown. Please discuss the rest of the data.

Response 2: We thank the reviewer for pointing out this deficit. The method of sensitivity test is not clear and has been modified. It was diluted for 3 times with ten fold serial diluted, respectively 1/10, 1/100 and 1/1000 of a J2 and cyst.

Point 3: Application to soil: This section needs much more information. Include soil types for each location. What was the population density of cysts/J2 in these samples? Where there cysts and J2 in all of the samples? How DNA was extracted from soil and volume of soil used must be included and described in detail. Mention of whether all three replicates resulted in the same result needs to be mentioned. If the opportunity exists, then the replication should be increased to 10 to demonstrate reproducibility of the method soil samples – this will make the results much stronger.

Response 3: We thank the reviewer for their careful consideration of our manuscript and for pointing out this deficit. As suggested, we have made many changes in this part. First of all, in the part of materials and methods, add the amount of soil and extracted detail used in the DNA extraction of nematode from soil. Secondly, the density of cysts in each soil sample is shown in table 3 of results. However, the number of J2 was not counted at that time. The results of three repeated tests are consistent. Unfortunately, the samples in this test have been destroyed and cannot be retested again.

Point 4: In the results and in a figure mention results from D2A/D3B – however, this is not mentioned in the methods. Make sure the reader knows these primers were used and why.

Response 4: The reason for PCR with the universal primer D2A/D3B of nematodes has been supplemented in the experimental method to confirm that the nucleic acid of nematodes has been successfully extracted from each sample.

Point 5: Why is the conclusion before the discussion?

Response 5: We are grateful to the reviewer for this suggestion.We have made changes and put the discussion part before the conclusion

Point 6: Statement comparing difficulty of using ITS vs. effector gene needs more context in discussion.

Response 6: We thank the reviewer for their careful consideration of our manuscript and for pointing out this deficit. This part of the discussion in the revised version has been deleted.

Point 7: It is state in discussion that cysts and J2 were in soil samples, but no evidence of this in methods or results.

Response 7:We are grateful to the reviewer for this suggestion. This part of the discussion in the revised manuscript has been modified. And add the density of cysts in each soil sample in the results.

Reviewer 2 Report

Dear Authors,

You introduced the LPA-LFA method to one of the most important pathogens of potato fields in the world. This method was shown in your paper to be effective, quick, and novel to this nematode and the results are impressive. I think your research can be greatly appreciated by researchers who need to identify PCN in a shorter time and have minimum laboratory equipment. However, I think the paper can be improved after covering/addressing some drawbacks.

Here are my main concerns about the manuscript.

-        I believe the English language in the manuscript would be a bit hard to follow for many readers. I recommend a careful revision of the whole manuscript.

-        You barely talk about LPA and nothing about LFA in the introduction section meanwhile some readers may be interested to know more about the method as you used it in the title and keywords.

-        The introduction is lacking the significance of the current work. The drawback of conventional molecular methods is not explained, and I see a poorly described background.

-         The “materials and methods” section is not well-defined, lacks important details, and can be confusing for researchers especially those who are less familiar with the method. I strongly recommend reorganizing this section and adding needed details.

-        If you examined the soil samples for PCN infestations in “Detection of Globodera rostochiensis from actual field soil”, You need to add them to the materials and methods and results section in brief.

You may also find the file enclosed containing detailed corrections and suggestions. I left some corrections/suggestions regarding the above-mentioned concerns. The important parts are highlighted in red and yellow indicating corrections and suggestions/questions, respectively.

I hope these suggestions can have a minor positive impact on improving your valuable work.

Cheers,

Reviewer

Author Response

Response to Reviewer 2 Comments

Thank you very much for your detailed corrections and suggestions. In the revised manuscript, we have modified all corrections and suggestions marked red and yellow. Our responses are marked in red, and the associated changes have been highlighted in the revised manuscript.

Your suggestion has played a great positive impact on our work, and we must once again express our most sincere thanks for your review.

Point 1: I believe the English language in the manuscript would be a bit hard to follow for many readers. I recommend a careful revision of the whole manuscript. 

Response 1: We thank the reviewer for his/her careful consideration of our manuscript and for pointing out this deficit. We carefully revised the whole manuscript.

Point 2:  You barely talk about LPA and nothing about LFA in the introduction section meanwhile some readers may be interested to know more about the method as you used it in the title and keywords.

Response 2: We are grateful to the reviewer for this suggestion. We have supplemented the content in the introduced section.

Point 3: The introduction is lacking the significance of the current work. The drawback of conventional molecular methods is not explained, and I see a poorly described background.

Response 3: We thank the reviewer for their careful consideration of our manuscript and for pointing out this deficit. We have removed “so far” and add the disadvantage of conventional molecular methods and several background.

Point 4:  The “materials and methods” section is not well-defined, lacks important details, and can be confusing for researchers especially those who are less familiar with the method. I strongly recommend reorganizing this section and adding needed details.

Response 4: We are grateful to the reviewer for this suggestion. We have mainly revised the “materials and methods”, and added the details of the extraction DNA from soil.

Point 5:  If you examined the soil samples for PCN infestations in “Detection of Globodera rostochiensis from actual field soil”, You need to add them to the materials and methods and results section in brief.

Response 5: We thank the reviewer for their careful consideration of our manuscript and for pointing out this deficit. As suggested, we have made many changes in this part. First of all, in the part of materials and methods, add the amount of soil and extracted detail used in the DNA extraction of nematode from soil. Secondly, the density of cysts in each soil sample is shown in table 3 of results. The results of three repeated tests are consistent. Unfortunately, the samples in this test have been destroyed and cannot be retested again.

Round 2

Reviewer 1 Report

The revised version of the manuscript is acceptable.